# Novel Uses of Al$_2$O$_3$/Mos$_2$ Hybrid Nanofluid in MQCL Hard Milling of Hardox 500 Steel

**Tran Minh Duc** [ID], **Tran The Long** *[ID] **and Ngo Minh Tuan** [ID]

Department of Manufacturing Engineering, Faculty of Mechanical Engineering, Thai Nguyen University of Technology, Thai Nguyen 250000, Vietnam; minhduc@tnut.edu.vn (T.M.D.); minhtuanngo@tnut.edu.vn (N.M.T.)
* Correspondence: tranthelong90@gmail.com or tranthelong@tnut.edu.vn; Tel.: +84-985-288-777

**Abstract:** In recent years, the application of environmentally friendly cutting fluids in the metal cutting industry has been a growing concern in all over the world. In this study, the minimum quantity cooling lubrication (MQCL) technique, which uses very small amount of cutting oil, is motivated to apply to the hard milling process of Hardox 500 steel. Further, rice bran oil, a natural biodegradable oil, is used as the base fluid of Al$_2$O$_3$/MoS$_2$ hybrid nanofluid. ANOVA analysis is used to study the influences of nanoparticle concentration, cutting speed, and feed rate on surface roughness. The obtained results indicate that good surface quality is achieved and the cutting speed is significantly increased to 140 m/min (about 2.55–2.80 times higher than the recommended values) due to the better cooling and lubricating effects from MQCL system and Al$_2$O$_3$/MoS$_2$ hybrid nanofluid. Moreover, the microstructure of the machined surface proves the formation of MoS$_2$ tribo film by using Al$_2$O$_3$/MoS$_2$ hybrid nanofluid, indicating that the effectiveness of each type of nanoparticle in hybrid nanofluid has been promoted. Furthermore, the important technical guides for machining Hardox 500 steel are provided.

**Keywords:** hard milling; MQCL; nanopartilces; nanofluid; Al$_2$O$_3$/MoS$_2$ hybrid nanofluid; difficult-to-cut material

## 1. Introduction

In recent years, there has been an increasing interest in sustainable machining, green manufacturing, and environmentally friendly manufacturing, which promote the research and application of new technologies to minimize negative impacts on the environment. The field of metal cutting is also not out of this trend. In the last two decades, hard machining technology, such as hard milling, has been successfully applied, bringing out economic, technological, environmental characteristics. This technology has attracted the attention not only by researchers but also by manufacturers all over the world. Especially in the mold industry, it is now possible to directly machine hardened steel with superior productivity compared to grinding and Electrical discharge machining (EDM) while still remaining technical requirements for dimensional accuracy and surface quality.

Hard milling is a method of directly processing steels after heat treatment, with high hardness (usually 45 HRC or more) [1,2]. During the hard milling process, the heat generated from the cutting zone is very large, which is the major disadvantage of this technology, as the large amount of heat generated accelerates cutting tool wear. Therefore, the cutting tool materials having high hardness and good heat resistance are required as coated carbide [3–6], ceramics [7], CBN [8], and diamond-coated inserts [9]. This causes the tool cost to be pushed up. To overcome this problem, it is necessary to take measures to reduce friction in the cutting zone, thereby reducing heat and improving the efficiency of the cutting process. In addition to that, metal cutting industries use flood cooling frequently, but the cutting process during milling is not continuous, so the use of the traditional flood coolant condition easily leads the thermal shock and causes tool chipping or tool failure. It greatly affects the tool life and the quality of the machined surface [5,6]. The initial solution

is hard milling under dry condition, which also contributes to maintain the environmental friendliness of hard milling technology, but the heat generated from the cutting zone is very large, speeding up the wear rate, and reducing tool life and the quality of machined surface. This not only limits the cutting conditions as well as the productivity, but also contributes to increase machining costs. In order to overcome these problems, there have been a number of studies that successfully applied minimum quantity lubrication (MQL) to hard milling. MQL is a method of directly spraying cutting fluid with high-pressure air flow into the cutting zone, thus providing superior lubrication compared to flood condition. There have been many studies showing the lubrication efficiency of MQL through the evaluation of the efficiency of the machining process in terms of cutting forces [10], cutting heat, surface quality, and so on [11,12]. L.N. Lopez de Lacalle et al. [13] performed a study on the effect of spraying cutting fluids of flood coolant and MQL conditions in high-speed milling. In this study, the position of the MQL nozzles with relation to feed direction, tool wear, cutting fluid consumption, and numerical simulation were investigated to point out that the cutting fluid under flood condition was not able to reach the inner zone of the tool teeth, but that under MQL condition can penetrate in cutting zone to help cool and lubricate the contact surfaces as well as remove the chips. The nozzle position also plays an important role in the effeciency of fluid spray.

In addition, due to the use of a very small amount of cutting fluid, MQL is considered an environmentally friendly technology, which will be a promising development alternative to wet and dry machining. The quantity of cutting fluids used in MQL technique is approximately 95% lower than that under flood coolant [13]. Together with the use of natural biodegradable oils as an alternative to mineral oil, this helps MQL technology not only improve the machining efficiency such as reducing cutting forces, tool wear, improving tool life but also retaining the properties of environmental friendliness [14], and this is technology is very suitable for the trend of sustainable development today.

However, the heat generated from hard milling process is very large, but the cooling efficiency of MQL is low, so the application of MQL in hard milling is still very limited, especially for difficult-to-cut materials like hardened steel, Ni alloy, and so on [13]. Hence, the choice of the cutting tools and cooling and lubricating condition play the very important roles in hard milling. Some novel promising approaches have been studied and proposed like $CO_2$ cryogenic, MQL using nano cutting fluids, or the combination of them.

O. Pereira and his co-authors [15] studied cutting forces and tool wear in $CO_2$ cryogenic combined with MQL milling of Inconel 718. The obtained results claimed that the cutting forces reduced about 22% and tool life incresed about 57% compared to MQL alone. Moreover, tool life prolonged about 40% and 65% when compared with external and internal $CO_2$ cryogenic cooling due to better lubricating performance. The authors also studied deeply the use of external and internal $CO_2$ cryogenic combined with MQL (CryoMQL) to be a suitable alternative for wet machining [16,17]. Furthermore, they found out that, from microstructure analysis, the thickness of the deformed layer under dry turning is much bigger than that under CryoMQL technique. It is proven that the CryoMQL technique provided superior cooling and lubricating effects compared to dry condition. The change of sub-surface microhardness under CryoMQL technique was also smaller than that of dry machining [18]. A.Rodríguez et al. [19] used liquefied $CO_2$ as cutting fluid for drilling process of CFRP-Ti6Al4V aeronautical stacks. The results reveal that the values of hole diameter diverged below 0.5% from nominal values, the cutting temperature much reduced, hole surface quality improved, and tool life significantly extended when compared with dry drilling.

At the same time, the application of nano cutting fluids as the base fluid of MQL technology used in hard machining has been proven to be an alternative solution for dry condition [20]. Nano cutting fluid is formed by mixing different types of nanoparticles such as $Al_2O_3$, $SiO_2$, $CuO$, $MoS_2$, $TiO_2$, CNT, and so on at a reasonable ratio. The purpose is to improve lubricating and cooling properties of the base fluids. In addition, the presence of nanoparticles in the solution also contributes to improving viscosity affecting on contact

line motion and dynamic wetting [21]. Recent studies have shown the effectiveness of using nano cutting fluid in MQL technology for machining difficult-to-cut materials. Gyoung-Ja Lee et al. [22] concluded that diamond nanofluid contributes to reduce the coefficient of friction by 23 wt.%, which leads to improve cutting efficiency and surface quality as well as reduce cutting temperature compared to the pure fluid. M.K. Ahmed Ali et al. [23] applied nano $Al_2O_3$ and $TiO_2$ lubricant in automotive engines with different mixing ratios. The experimental results showed that the coefficient of friction and wear decreased by 11 wt.% and 2.6 wt.%, respectively. When analyzing the wear land, the authors found that the presence of $Al_2O_3$ nanoparticles in lubricating oil contributes to improve wear resistance due to the formation of a protective thin layer on the part surface. Meanwhile, $TiO_2$ nano fluid is effective in reducing the coefficient of friction. M.K.Sinha and his co-authors [24] studied and applied ZnO vegetable-based nanofluid with MQL technology for grinding Inconel 718 alloy, a difficult-to-cut material. The results of the study showed that the shear force, the coefficient of friction reduced, and the surface quality improved due to the improvement in the lubrication performance in cutting zone even when the cutting temperature was higher than that of flood condition. A.K. Sharma et al. [25] has conducted an overview study of the efficiency of MQL technology applied to machining processes using traditional pure fluids and nanofluids. The author pointed out that the effectiveness of the application of MQL with nanofluid in milling process significantly reduced cutting forces, cutting temperature, tool wear, and lubricant usage, and also enhanced the machined surface quality. N.A.C. Sidik et al. [26] also carried out an overview study of the nanofluid application for MQL technology in metal cutting processes. The study results also showed the improvement in the friction, abrasion, and lubrication properties as well as the coefficient of thermal conductivity. However, the author also recommends that it is necessary to have follow-up studies to further discover and optimize the parameters of the nanofluids, thereby maximizing the efficiency of the cutting process.

Effective application of nanofluids requires researching and investigating some basic parameters such as base fluid type, type of nanoparticle, nanoparticle concentration, of nanoparticle size, and morphology of nanoparticles. Pasam et al. [27] studied the effect of MQL using the base fluid containing nanoparticles and microparticles on cutting AISI 1040 steel. The obtained results indicated that the use of fluids containing microparticles gives higher economic efficiency in rough cutting, but for finishing, the use of nanofluids will be better. Hegab et al. [28] studied the effect of CNT nanofluid in MQL turning of Ti-6Al-4V alloy and concluded that the surface quality improved and the tool wear reduced due to the improvement in lubricating and cooling properties of nanofluid. Nanoparticle concentration and feed rate are two parameters that greatly affect the machined surface quality. Therefore, the authors also conducted the research on the machinability of the tool and the chip morphology during turning Inconel 718 alloy with MQL technology using CNT and $Al_2O_3$ nanofluids [29,30]. The study results indicated that the tool machinability improved, and the chip thickness reduced due to better cooling and lubricating performance of nanofluids. However, it is necessary to have more studies on the parameters of nano cutting fluids so that it can be effectively and widely applied in different machining methods.

In recent years, to meet the increasing demand for productivity and quality in metal cutting industry, especially in the field of processing difficult-to-cut materials. This raises new requirements for lubrication and cooling in the cutting zone but still meets new strict environmental standards, especially for heavy machining methods generating enormous amount of heat like hard milling and grinding. Accordingly, a new trend in nano cutting fluid applications is to use a combination of two different types of nanoparticles suspended in the base fluid to form a hybrid nanofluid. Such use aims to take advantage of the most outstanding properties of each type of nanoparticle, thereby further improving the lubricating and cooling efficiency of the base fluid [31].

There have been a number of publications showing the effectiveness of the application of hybrid nanofluids in machining, but mainly focused on the turning process with

MQL technique [31–33], while the studies on hard milling process with minimum quantity cooling lubrication (MQCL) technique is very little information [34–36]. Therefore, the author conducted a study on the effect of $Al_2O_3/MoS_2$ hybrid nanofluid on surface roughness in hard milling of 500 Hardox steel under MQCL condition. This type of steel produced by SSAB company, SWEDEN [37]. Commercial hardox steel has been completely heat-treatment by the supplier, so it has the fairly high hardness, high strength, and also high ductility as well as good wear resistance. Hence, Hardox 500 steel is widely used in industrial practice and is grouped among the difficult-to-cut materials. Therefore, the successful application of MQCL using $Al_2O_3/MoS_2$ hybrid nanofluid to improve the cutting efficiency of hard milling process play an important role in technical, economic, and environmental characteristics. In this work, the investigation mainly focuses on the effects of nanoparticle concentration, cutting speed, and feed rate on surface roughness and surface microstructure.

## 2. Material and Method

### 2.1. Experimental Set Up

The design of experiment is shown in Figure 1. In this study, Mazak vertical center smart 530C (Yamazaki Mazak Corporation, 1-131 Takeda, Oguchi-cho, Niwa-gun, Aichi-Pref, Japan) was used to conduct the experiments. Face mill head with the designation of SHIJIE BAP 400R-50-22-4T was used. The TiAlN coated carbide inserts (LAMINA TECHNOLOGIES SA, Yverdon-les-Bains, Switzerland, Available online: https://wix.laminatech.ch/img/catalog/1237.pdf (accessed on 5 June 2019)) are APMT 1604 PDTR LT30 PVD.

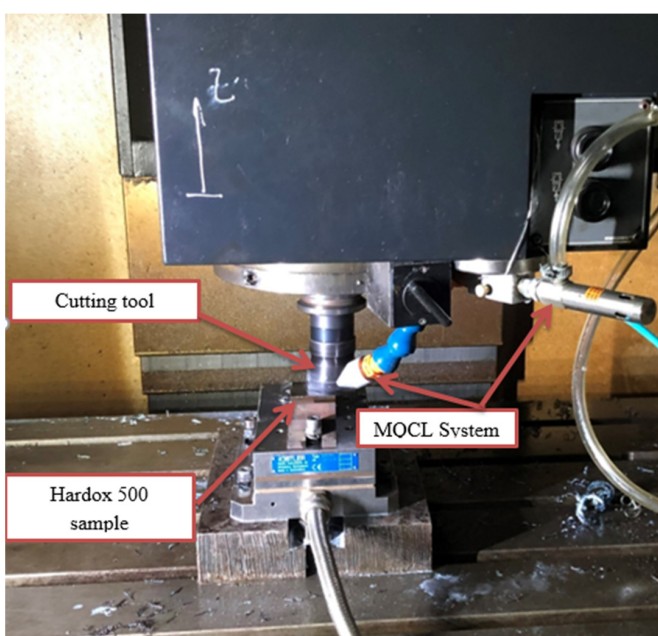

**Figure 1.** The experimental set up.

The MQCL nozzle named Frigid-X Sub-Zero Vortex Tool Cooling Mist System (made by Nex Flow™, Richmond Hill, ON, Canada) was used with the rice bran oil based-water-based $Al_2O_3/MoS_2$ hybrid nanofluid. $Al_2O_3$ and $MoS_2$ nanoparticles were made by Soochow Hengqiu Graphene Technology Co., Ltd. (Suzhou, China) and Luoyang Tongrun Info Technology Co., Ltd. (Luoyang, China) with the size of 30 nm (average), respectively. To ensure the uniform suspension of $Al_2O_3$ and $MoS_2$ nanoparticles with mixing ratio of 90:10 in rice bran oil, Ultrasons-HD ultrasonicator (JP SELECTA, Abrera, Spain) generating 600 W ultrasonic pulses at 40 kHz was used for 1 h and the obtained $Al_2O_3/MoS_2$ hybrid nanofluid was directly used for MQCL system.

SJ-210 Mitutoyo (Mitutoyo Corporation, Kawasaki, Kanagawa, Japan) was used for measuring the surface roughness. KEYENCE VHX-6000 Digital Microscope (KEYENCE Corporation, Osaka, Japan) was used to investigate the surface topography. In this research, the workpiece samples of Hardox 500 steels with the dimensions of 150 ×100 ×15 mm were used.

### 2.2. Experiment Design

Minitab 18.0 software (3rd floor, N03-T5 Embassy Garden, Vo Chi Cong Street, Xuan Tao Ward, Bac Tu Liem District, Ha Noi, Vietnam) is used for the Box-Behnken experimental design with three input machining variables and their levels are listed in Table 1. The levels of nanoparticle concentration and feed rate were selected based on previous studies [37]. The cutting speed chosen with such a large interval is due to the following reasons. First, Hardox 500 steel is a difficult-to-cut material because it possesses both high hardness and ductility properties at the same time. Therefore, the cutting speed value of 80 m/min is chosen according to the manufacturer's recommendation and the cutting speed of 140 m/min is chosen for the purpose of investigating the cooling lubrication efficiency of MQCL technique using $Al_2O_3/MoS_2$ hybrid nanofluid as well as observing how much improvement for productivity it is.

**Table 1.** Input machining variables and their levels.

| Input Machining Variables | Unit | Symbol | Level | |
|---|---|---|---|---|
| | | | Low | High |
| Nanoparticle concentration | wt.% | $NC$ | 0.5 | 1.5 |
| Cutting speed | m/min | $V_c$ | 80 | 140 |
| Feed rate | mm/tooth | $F$ | 0.08 | 0.16 |

Table 2 summarizes the experiment design with test run order and the measured values of surface roughness. The fixed parameters are the depth of cut of 0.12 mm, air pressure of 0.6 MPa, flow rate of 30 mL/h. Each of the trials are repeated three times under the same cutting condition and the average values are reported.

**Table 2.** The experiment design with test run order and the measured values of surface roughness.

| Std Order | Run Order | PtType | Blocks | Input Machining Variables | | | Response Variables |
|---|---|---|---|---|---|---|---|
| | | | | $NC$ (wt.%) | $Vc$ (m/min) | $F$ (mm/tooth) | $R_a$ (μm) |
| 1 | 10 | 2 | 1 | 0.5 | 80 | 0.12 | 0.162 |
| 2 | 2 | 2 | 1 | 1.5 | 80 | 0.12 | 0.177 |
| 3 | 1 | 2 | 1 | 0.5 | 140 | 0.12 | 0.136 |
| 4 | 18 | 2 | 1 | 1.5 | 140 | 0.12 | 0.126 |
| 5 | 16 | 2 | 1 | 0.5 | 110 | 0.08 | 0.118 |
| 6 | 22 | 2 | 1 | 1.5 | 110 | 0.08 | 0.099 |
| 7 | 4 | 2 | 1 | 0.5 | 110 | 0.16 | 0.156 |
| 8 | 24 | 2 | 1 | 1.5 | 110 | 0.16 | 0.182 |
| 9 | 30 | 2 | 1 | 1 | 80 | 0.08 | 0.129 |
| 10 | 12 | 2 | 1 | 1 | 140 | 0.08 | 0.085 |
| 11 | 29 | 2 | 1 | 1 | 80 | 0.16 | 0.196 |
| 12 | 27 | 2 | 1 | 1 | 140 | 0.16 | 0.173 |
| 13 | 15 | 0 | 1 | 1 | 110 | 0.12 | 0.174 |
| 14 | 6 | 0 | 1 | 1 | 110 | 0.12 | 0.157 |
| 15 | 11 | 0 | 1 | 1 | 110 | 0.12 | 0.166 |
| 16 | 14 | 2 | 1 | 0.5 | 80 | 0.12 | 0.167 |
| 17 | 5 | 2 | 1 | 1.5 | 80 | 0.12 | 0.146 |
| 18 | 28 | 2 | 1 | 0.5 | 140 | 0.12 | 0.159 |

**Table 2.** *Cont.*

| Std Order | Run Order | PtType | Blocks | Input Machining Variables | | | Response Variables |
|---|---|---|---|---|---|---|---|
| | | | | NC (wt.%) | Vc (m/min) | F (mm/tooth) | $R_a$ (µm) |
| 19 | 9 | 2 | 1 | 1.5 | 140 | 0.12 | 0.124 |
| 20 | 13 | 2 | 1 | 0.5 | 110 | 0.08 | 0.137 |
| 21 | 23 | 2 | 1 | 1.5 | 110 | 0.08 | 0.086 |
| 22 | 3 | 2 | 1 | 0.5 | 110 | 0.16 | 0.201 |
| 23 | 21 | 2 | 1 | 1.5 | 110 | 0.16 | 0.181 |
| 24 | 19 | 2 | 1 | 1 | 80 | 0.08 | 0.127 |
| 25 | 8 | 2 | 1 | 1 | 140 | 0.08 | 0.084 |
| 26 | 7 | 2 | 1 | 1 | 80 | 0.16 | 0.153 |
| 27 | 25 | 2 | 1 | 1 | 140 | 0.16 | 0.160 |
| 28 | 20 | 0 | 1 | 1 | 110 | 0.12 | 0.176 |
| 29 | 26 | 0 | 1 | 1 | 110 | 0.12 | 0.159 |
| 30 | 17 | 0 | 1 | 1 | 110 | 0.12 | 0.152 |

## 3. Results and Discussion

### 3.1. The Effects of Input Machining Variables on Surface Roughness

The ANOVA analysis with 95% confidence level is carried out, and the regression model of surface roughness $R_a$ with $R^2$ equal to 87.76% is given below in Equation (1). Table A1 (Appendix A) shows the result of ANOVA analysis.

$$R_a = -0.048 - 0.0046 * NC + 0.00157 * V_c + 1.819 * F - 0.0155 * NC * NC$$
$$- 0.000012 * V_c * V_c - 9.45 * F * F - 0.000325 * NC * V_c \quad (1)$$
$$+ 0.475 * NC * F + 0.00740 * V_c * F$$

The Pareto chart of the standardized effects with $\alpha = 0.05$ for the output variable $R_a$, exhibits the effects of the input machining factors shown in Figure 2. Feed rate (*F*) has the strongest influence on $R_a$, followed by cutting speed ($V_c$) and nanoparticle concentration (*NC*). The surface roughness value reflects the machined surface profile. These micro peaks and valleys are the result of the cutting tool scratching over the machined surface. Hence, as the toolpath increases, the spacing between peaks increases. In addition, Hardox sheet also has extra-high toughness, which greatly influences on the plastic deformation of the machined surface. Accordingly, when the toolpath increases, the plastic deformation increases, so the surface roughness go up. That is why the amount of feed rate has the greatest influence on the surface roughness value among the input cutting conditions.

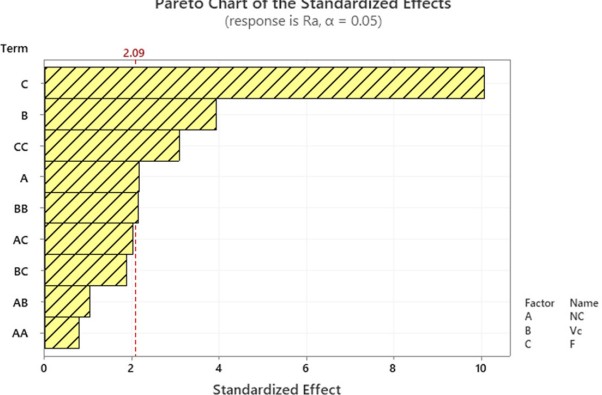

**Figure 2.** Pareto chart of effects of input machining factors on surface roughness $R_a$. (A is *NC*: nanoparticle concentration, B is $V_c$: cutting speed, C is *F*: feed rate, AA is the quadratic effect of nanoparticle concentration, BB is the quadratic effect of cutting speed, and CC is the quadratic effect of feed rate).

The quadratic effect CC (*FF*) shows the significant influence on the investigated function of $R_a$, followed by the quadratic effect BB ($V_cV_c$). The other quadratic effects and two-way interaction effects exhibit very little influence on $R_a$ (Figure 3).

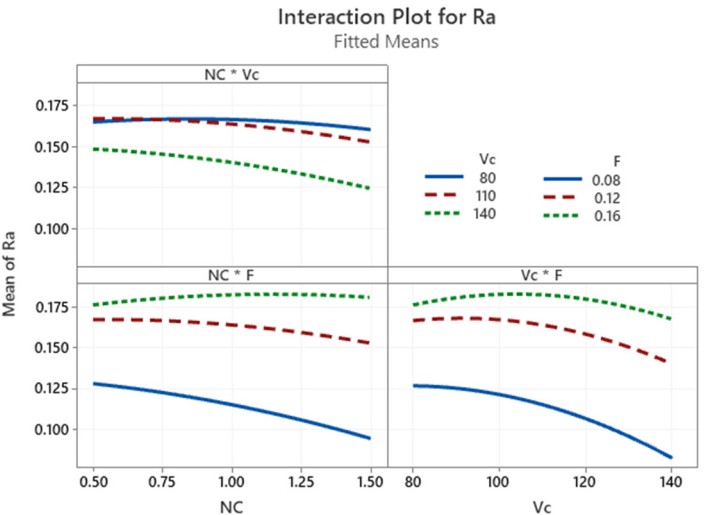

**Figure 3.** Interaction plot of input machining factors on surface roughness $R_a$.

Figure 4 shows the plot of main effects of input machining factors on surface roughness $R_a$. It can be seen that the feed rate has a great influence on the surface roughness values. As the feed rate increases, the roughness values go up rapidly. Meanwhile, increasing the nanoparticle concentration value and cutting speed helps to reduce the surface roughness.

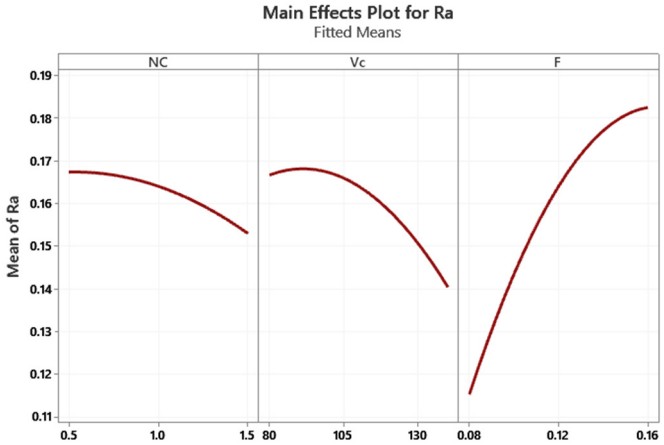

**Figure 4.** Main effects plot of input machining factors on surface roughness $R_a$.

From the residual plots (Figure 5), it can be seen that the Normal Probability Plot compares the probability of distribution of the residual values displayed in points with the normal distribution displayed as a straight line. The residual values fit well to the normal line. The histogram graph shows that the frequency of residual values centered around the center of distribution, which can be considered according to the normal distribution law. The versus fit graph represents the relationship between the residuals and their respective values of the regression model. These points are distributed very randomly around the 0 line, which proves that the imported $R_a$ data is not affected by any rule control factors other than the input variables. The versus order graph represents the relationship between the residuals and the order of data points. These points are distributed randomly around the 0 line, which proves whether the imported Ra is not affected by the time factor.

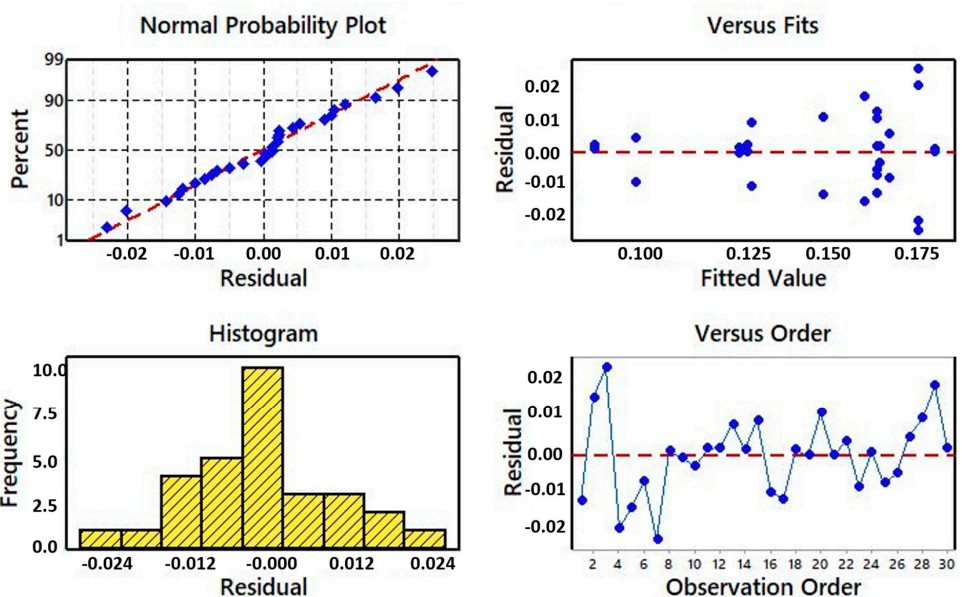

**Figure 5.** Residual plots of surface roughness $R_a$.

Surface and contour plots of the influence of the investigated variables on the surface roughness values $R_a$ are shown in Figures 6–8. For holding the feed rate of 0.12 mm/tooth, it can be seen that using a high nano concentration of 1.5 wt.% and a large cutting speed of 140 m/min gives the minimum surface roughness (Figure 6). When keeping the cutting speed of 110 m/min, a high nanoparticle concentration of 1.5 wt.% and a small feed rate of 0.08 mm/tooth should be used for the minimum surface roughness (Figure 7). It can be explained that the cutting speed is higher than the rate of plastic deformation on the machined surface. Hence, the plastic deformation reduces, so the surface roughness value decreases. On the other hand, using a high nanoparticle concentration means more nanoparticles participate in the cutting zone, in which $MoS_2$ thin film is formed and the roller mechanism of $Al_2O_3$ nanoparticles is accelerated when increasing the cutting speed. Accordingly, the lubricating effect is promoted, and the surface quality improves. When keeping the nanoparticle concentration of 1.0 wt.%, the use of a large cutting speed of 140 m/min and a small feed rate of 0.08 mm/tooth bring out the minimum values of surface roughness (Figure 8).

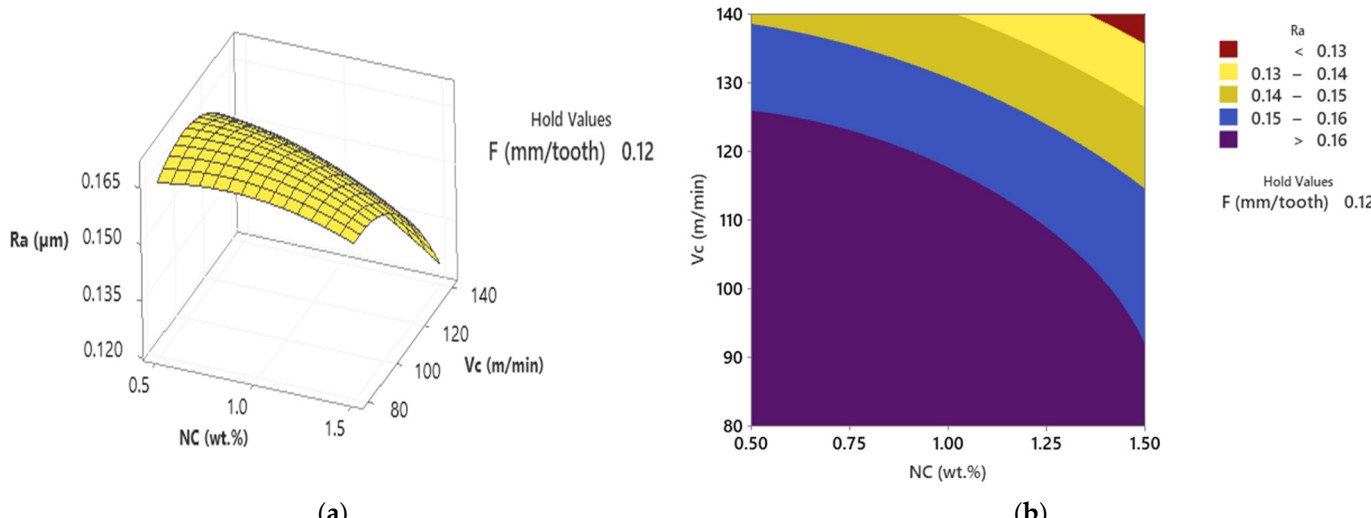

(**a**)　　　　　　　　　　　　　　　　　　　　　　　　　　　　　　　　(**b**)

**Figure 6.** Effects of nanoparticle concentration and cutting speed on surface roughness $R_a$: (**a**) surface plot, (**b**) contour plot.

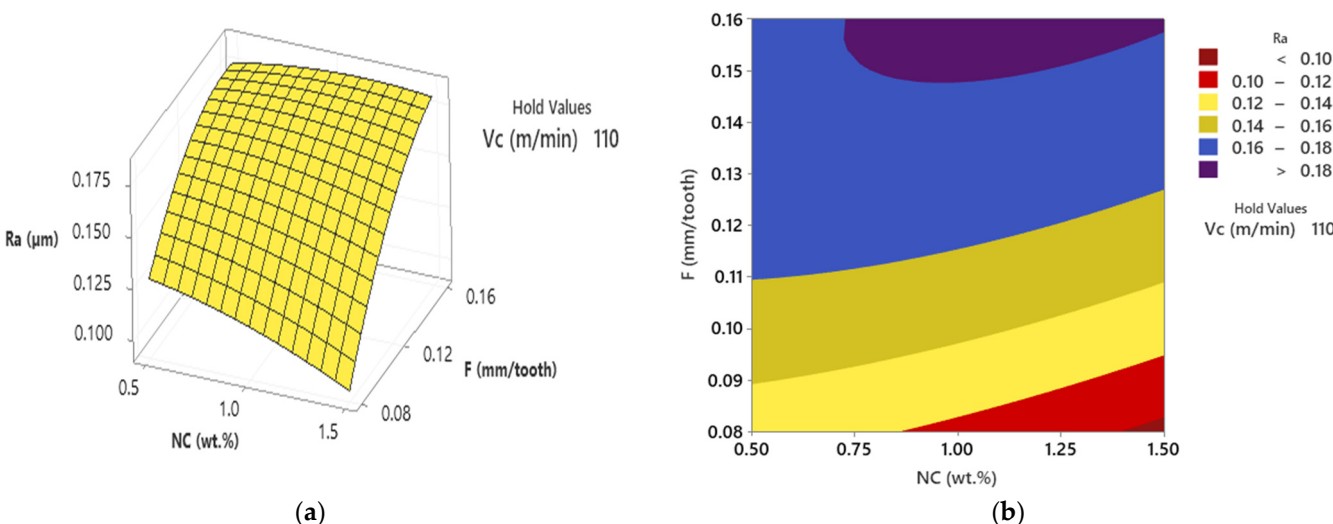

**(a)**                                                    **(b)**

**Figure 7.** Effects of nanoparticle concentration and feed rate on surface roughness $R_a$: (**a**) surface plot, (**b**) contour plot.

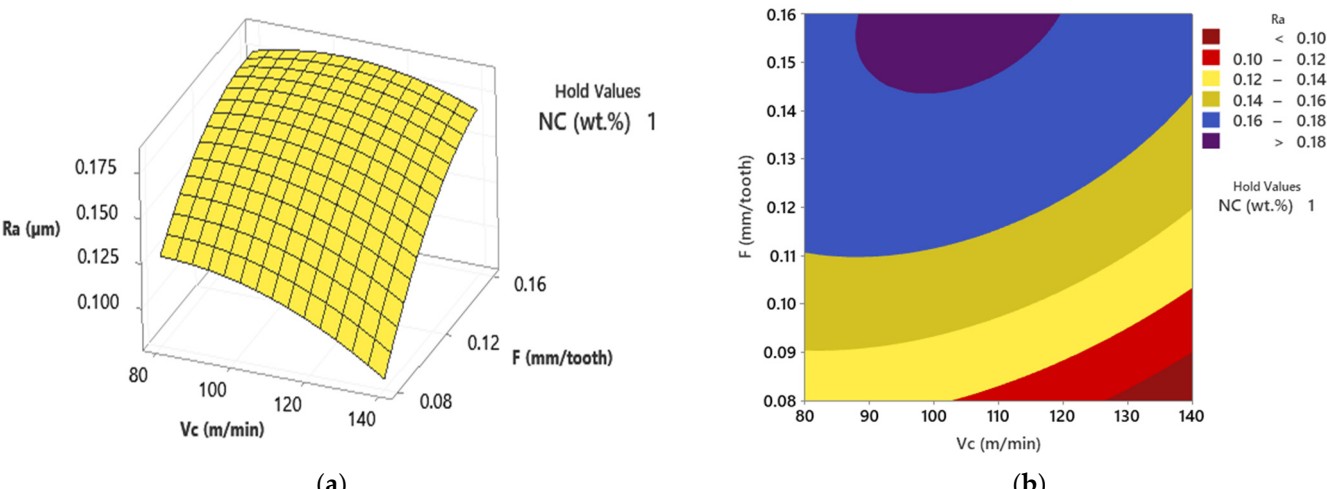

**(a)**                                                    **(b)**

**Figure 8.** Effects of cutting speed and feed rate on surface roughness $R_a$: (**a**) surface plot, (**b**) contour plot.

Feed rate has the strongest influence on surface roughness. As it increases, $R_a$ values go up sharply. This result is also consistent with the general rule of the impact of feed rate on other cutting machining methods. The concentration of nanoparticles has a great influence on the machined surface quality. The surface quality improves with the increase of nanoparticle concentration. The cutting speed has a little effect on the roughness $R_a$. When it increases, $R_a$ values decrease.

In $Al_2O_3/MoS_2$ hybrid nanofluid, $Al_2O_3$ nanoparticles contribute to increase the thermal conductivity of the base fluid and these particles play an important role in creating ball roller effect [19], while $MoS_2$ nanoparticles have the effect of increasing viscosity and forming tribo film, which reduces the friction in the cutting zone [35,38]. In the investigated range of nanoparticle concentration, when the nano concentration of hybrid nanofluid increases, the concentration of $MoS_2$ nanoparticles also rises, thus increasing the ability to create the tribo film on the machined surface. In detail, for the concentration of the hybrid nanofluid of 1.5 wt.%, the corresponding $MoS_2$ nanoparticle concentration is 0.15 wt.%, so the creation of a "thin film" on the surface is the most obvious and agrees with the previous study [35].

### 3.2. Investigation of Surface Microstructure

Figure 9 shows the microstructure of the machined surface with the MQCL condition using two different types of base fluids including: (a) 1.5 wt.% $Al_2O_3$ nanofluid, (b) 1.5 wt.% $Al_2O_3/MoS_2$ hybrid nanofluid. It can be clearly seen that by using 1.5 wt.% $Al_2O_3/MoS_2$ hybrid nanofluid, there exists a "thin bubble film" on the machined surface due to the effect of $MoS_2$ nanoparticles in Figure 9b, while this layer is absent when using 1.5 wt.% $Al_2O_3$ nanofluid as shown in Figure 9a. In this study, when the concentration of the hybrid nanofluid increases to 1.5 wt.%, the corresponding $MoS_2$ nanoparticle concentration will be 0.3 wt.%. Since the concentration of $MoS_2$ particles of 0.15 wt.% is not the optimal concentration [35], as shown in Figures 6 and 7, the surface roughness graphs continue to decrease with the rise of nano concentration. This result is consistent with the previous studies [35] because when the concentration of $MoS_2$ nanoparticles increases to over 0.5 wt.%, it will adversely affect the machined surface [35,37].

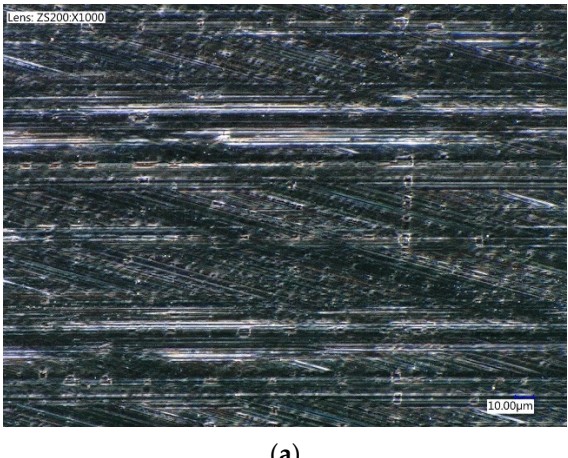
(**a**)

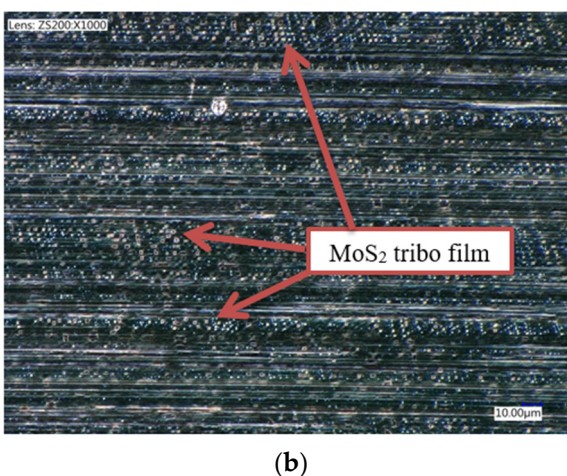
(**b**)

**Figure 9.** Microstructure of machined surface with: (**a**) 1.5 wt.% $Al_2O_3$ nanofluid, (**b**) 1.5 wt.% $Al_2O_3/MoS_2$ hybrid nanofluid.

The effect of the nanoparticle concentration for using hybrid nanofluids is different from that using only $MoS_2$ nanofluid. $MoS_2$ nanoparticles are effective in improving lubricating performance, but they adversely affect the cutting process in case of using inappropriate concentration.

The use of $MoS_2$ nanoparticles in the $Al_2O_3/MoS_2$ hybrid nanofluid with the appropriate ratio will benefit from the advantages and limit the disadvantages of each type of nanoparticle. Specifically, the improvement of thermal conductivity and ball roller effects caused by $Al_2O_3$ nanoparticles as well as the lubrication ability of $MoS_2$ nanoparticles will be promoted with only a small ratio in the base fluids.

On the other hand, nanoparticles suspended in cutting fluids in oil mist form will be threats for human health, so the use of exhaust fans and a ventilation system are suggested to remove the oil mist when applying this technique.

### 4. Conclusions

In this study, the effect of the $Al_2O_3/MoS_2$ hybrid nanofluid with the base fluid of rice bran oil on cutting performance and surface roughness in hard milling of Hardox 500 steel was investigated by using ANOVA analysis applied for the Box-Behnken experimental design. The effects of nanoparticle concentration, cutting speed, and feed rate on surface roughness has been studied and evaluated. The initial obtained results show that, by using the $Al_2O_3/MoS_2$ hybrid nanofluid, the advantages of each type of nanoparticle have been promoted.

The surface and contour plots were formed to evaluate the interacting effects of input machining parameters on surface roughness. From these, the technical guides for machining Hardox 500 steel were provided and led to the research direction for further investigations.

The microstructure of machined surface under $Al_2O_3/MoS_2$ hybrid nanofluid MQCL condition was studied and compared to that under $Al_2O_3$ nanofluid MQCL condition. The formation of thin tribo film from $MoS_2$ nanosheet will be a very interesting finding. This finding will be supported by the trend of using hybrid nanofluids.

Furthermore, the successful application of MQCL technology using $Al_2O_3/MoS_2$ hybrid nanofluid for hard milling of Hardox 500 steel, a difficult-to-cut material, is a new solution, which can replace dry and wet machining. Experimental results have shown the outstanding lubrication and cooling effects which will contribute important technical information for the machining industry.

From the obtained results, the cutting speed can be risen from 80 m/min to 140 m/min under $Al_2O_3/MoS_2$ hybrid nanofluid MQCL condition to improve the productivity while retaining the good surface quality, which is a little bit better than that of grinding. Moreover, the cutting speed of 140 m/min can be effectively used for hard milling of Hardox 500 steel, which is about 2.55–2.80 times higher than those of manufacturer's recommendations [38], which bring out the economic and technological effectiveness.

From an environmental point of view, the application of MQCL, which uses a very small amount of cutting oil combined with rice bran oil, a type of natural biodegradable oil, reveals a promising solution for sustainable production.

In further work, more investigation is needed to focus on the effects of $Al_2O_3/MoS_2$ hybrid nanofluid on tool wear, tool life, and cutting temperature. In addition to this, the ratio of $Al_2O_3/MoS_2$ nanoparticles will be studied.

**Author Contributions:** Conceptualization, T.M.D. and T.T.L.; software, N.M.T., validation, N.M.T.; formal analysis, T.T.L. and N.M.T.; investigation, N.M.T.; data curation, T.T.L. and N.M.T.; writing—original draft preparation, T.M.D. and T.T.L.; writing—review and editing, T.T.L.; supervision, T.M.D. and T.T.L.; project administration, T.M.D. and T.T.L. All authors have read and agreed to the published version of the manuscript.

**Funding:** This research was funded by Thai Nguyen University of Technology, Thai Nguyen University, Vietnam with the project number of T2020-B03.

**Acknowledgments:** The work presented in this paper is supported by Thai Nguyen University of Technology, Thai Nguyen University, Vietnam.

**Conflicts of Interest:** The authors declare no conflict of interest.

**Appendix A**

**Table A1.** Results of the ANOVA analysis of surface roughness $R_a$.

| Source | DF | Adj SS | Adj MS | F-Value | *p*-Value |
|--------|----|--------|--------|---------|-----------|
| Model | 9 | 0.025516 | 0.002835 | 15.93 | 0.000 |
| Linear | 3 | 0.021606 | 0.007202 | 40.46 | 0.000 |
| $NC$ | 1 | 0.000827 | 0.000827 | 4.64 | 0.044 |
| $V_c$ | 1 | 0.002756 | 0.002756 | 15.49 | 0.001 |
| $F$ | 1 | 0.018023 | 0.018023 | 101.26 | 0.000 |
| Square | 3 | 0.002368 | 0.000789 | 4.44 | 0.015 |
| $NC*NC$ | 1 | 0.000111 | 0.000111 | 0.62 | 0.439 |
| $V_c*V_c$ | 1 | 0.000814 | 0.000814 | 4.57 | 0.045 |
| $F*F$ | 1 | 0.001689 | 0.001689 | 9.49 | 0.006 |

**Table A1.** *Cont.*

| Source | DF | Adj SS | Adj MS | F-Value | *p*-Value |
|---|---|---|---|---|---|
| *2-way interaction* | 3 | 0.001542 | 0.000514 | 2.89 | 0.061 |
| $NC*V_c$ | 1 | 0.000190 | 0.000190 | 1.07 | 0.314 |
| $NC*F$ | 1 | 0.000722 | 0.000722 | 4.06 | 0.058 |
| $V_c*F$ | 1 | 0.000630 | 0.000630 | 3.54 | 0.075 |
| Error | 20 | 0.003560 | 0.000178 | | |
| Lack-of-Fit | 3 | 0.000045 | 0.000015 | 0.07 | 0.974 |
| Pure Error | 17 | 0.003515 | 0.000207 | | |
| Total | 29 | 0.029076 | | | |

* represents the interactions between the factors.

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
