# Peer review of "Novel Uses of Al2O3/Mos2 Hybrid Nanofluid in MQCL Hard Milling of Hardox 500 Steel"

_lubricants, doi:10.3390/lubricants9040045_

Round 1

Reviewer 1 Report

The paper is dealing with study the effect of the Al2O3/MoS2 hybrid nanofluid with the base fluid of soybean oil on cutting performance and surface roughness in hard milling of Hardox 500 steel by using ANOVA analysis applied for the Box-Behnken experimental design. The effects of nanoparticle concentration, cutting speed, and feed rate on surface roughness had been studied and evaluated. The initial obtained results show that by using the Al2O3/MoS2 hybrid nanofluid, the advantages of each type of nanoparticle have been promoted. 

Some minor points have to be considered before publication:

The font of the paper is not the same, see page 4 and 9

The references are not in the same stile of the Lubricants.

The figures and tables are not centered.

Regarding the results and discussion:  I do not know the reason of the large intervals of Input machining variables and their levels, Table 3.

Author Response

RESPONSE TO THE REVIEWER 1

We are very grateful for the reviews provided by the editors and each of the external reviewers of this manuscript. Please see below, our detailed response to comments.

The paper is dealing with study the effect of the Al2O3/MoS2 hybrid nanofluid with the base fluid of soybean oil on cutting performance and surface roughness in hard milling of Hardox 500 steel by using ANOVA analysis applied for the Box-Behnken experimental design. The effects of nanoparticle concentration, cutting speed, and feed rate on surface roughness had been studied and evaluated. The initial obtained results show that by using the Al2O3/MoS2 hybrid nanofluid, the advantages of each type of nanoparticle have been promoted. 

Some minor points have to be considered before publication:

1.The font of the paper is not the same, see page 4 and 9

Answer:

The manuscript was revised carefully and modified by following your comments. Thank you very much.

  1. The references are not in the same stile of the Lubricants.

Answer:

The references were modified to the format of the Lubricants.

  1. The figures and tables are not centered.

Answer:

The manuscript was revised carefully and modified by following your comments. Thank you very much.

  1. Regarding the results and discussion:  I do not know the reason of the large intervals of Input machining variables and their levels, Table 3.

Answer:

Thank you for your valuable comment. The levels of nanoparticle concentration ​​and feedrate were selected based on previous studies [31]. The cutting speed chosen with such a large interval is due to the following reasons. First, Hardox 500 steel is a difficult-to-cut material because it possesses both high hardness and ductility properties at the same time. Therefore, the cutting speed value of 80 m/min is chosen according to the manufacturer's recommendation and the cutting speed of 140 m/min is chosen for the purpose of investigating the cooling lubrication efficiency of MQCL technique using Al2O3/MoS2 hybrid nanofluid as well as observing how much improvement for productivity it is.

The discussion for the reason was added in the revised manuscript

Reviewer 2 Report

1. Define MQCL 2. Please discuss threats for human health associated with the nanofluids application as cutting fluids? 3. Please do not use citations in single space. Only cite some relevant papers and mention contribution of each article (see introduction section) 4. In introduction section, add some scientific reasoning should be added 5. Industries uses flood cooling frequently. Please correct this statement at page 1 last paragraph 6. Quality of fig 2 and fig 3 is poor. I would recommend to use your own figures rather than citing others figures with poor quality 7. Please cite Table 1 and 2 if you did not check the composition and mechanical properties 8. Provide table of design of experiments in text not end of the paper. 9. why surface roughness is higher for experimental condition c 10. what is x1, x2, and x3? 11. Arrange fig 6 12. Formatting is poor. Font size is different 13. Extend discussion for Fig 4 14. Please cite Fig 8, 9 and enlarge discussion with discussion about physics involved behind lubrication 15. Quality of fig 11 is poor 16. Rewrite conclusion and abstract with addition of key points 17. surface roughness is not enough. Add tool wear or SEM and EDS of surface profile

Author Response

RESPONSE TO THE REVIEWER 2

We are very grateful for the reviews provided by the editors and each of the external reviewers of this manuscript. Please see below, our detailed response to comments.

  1. Define MQCL

Answer:

The term MQCL was defined in the revised manuscript

  1. Please discuss threats for human health associated with the nanofluids application as cutting fluids?

Answer:

Thank you very much for your valuable comment. We also encountered with this problem when doing the experiments. Nanoparticles suspended in cutting fluids in oil mist form will be threats for human health, so the exhaust fans and ventilation system mist are suggested to use to remove the oil mist. The discussion about this problem was added in the revised manuscript in Sec. 3. Discussion with red color.

In our workshop, we already had the exhaust fans and ventilation system to remove the oil mist in order to avoid this problem.

  1. Please do not use citations in single space. Only cite some relevant papers and mention contribution of each article (see introduction section)

Answer:

The manuscript was revised by following the reviewer’s comments in the introduction section

Therefore, the cutting tool materials having high hardness and good heat resistance are required as coated carbide [3-6], ceramics [7], CBN [8], and diamond-coated inserts [9].

  1. In introduction section, add some scientific reasoning should be added

Answer:

The manuscript was revised by following the reviewer’s comments by adding some relevant studies and scientific reasoning.

  1. Industries uses flood cooling frequently. Please correct this statement at page 1 last paragraph

Answer:

Thank you very much. The authors corrected this statement at last paragraph on page 1 in the revised manuscript

  1. Quality of fig 2 and fig 3 is poor. I would recommend to use your own figures rather than citing others figures with poor quality

Answer:

The Figs. 2,3 were removed in the revised manuscript to improve the paper quality because I do not use them to analyze.

  1. Please cite Table 1 and 2 if you did not check the composition and mechanical properties

Answer:

The Table 1 and 2 were removed and cited in the revised manuscript

  1. Provide table of design of experiments in text not end of the paper.

Answer:

The table of design of experiments was moved in the proper position in the revised manuscript

  1. why surface roughness is higher for experimental condition c

Answer:

Thank for your comment, but the authors did not understand “experimental condition c”. Please give us the question again.

  1. what is x1, x2, and x3?

Answer:

The variables x1, x2, and x3 were changed to the names of nanoparticle concentration, cutting speed, and feed rate. Thank you very much

  1. Arrange fig 6

Answer:

The Fig. 6 was moved in the proper position in the revised manuscript

  1. Formatting is poor. Font size is different

Answer:

Format was modified to the journal’s form, and Font size was revised carefully. Thank you

  1. Extend discussion for Fig 4

Answer:

The discussion of Fig.4 in Sec. 3.1 was extended in the revised manuscript with red color

  1. Please cite Fig 8, 9 and enlarge discussion with discussion about physics involved behind lubrication

Answer:

Figs. 8, 9 are cited and the discussion for them is enlarged in the revised manuscript with red color

It can be explained that the cutting speed is higher than the rate of plastic deformation on the machined surface. Hence, the plastic deformation reduces, so the surface roughness value decreases. On the other hand, using a high nanoparticle concentration means more nanoparticles participate in the cutting zone, in which MoS2 thin film is formed and the roller mechanism of Al2O3 nanoparticles is accelerated when increasing the cutting speed. Accordingly, the lubricating effect is promoted and the surface quality improves.

  1. Quality of fig 11 is poor

Answer:

The images of Fig. 11 were changed to the full-size ones to improve the quality.

  1. Rewrite conclusion and abstract with addition of key points

Answer:

The conclusion and abstract were rewritten with addition of key points by following the reviewer’s comments. The change is put in red color.

  1. surface roughness is not enough. Add tool wear or SEM and EDS of surface profile

Answer:

The results of cutting forces, tool wear and tool life are being investigated, and they will be analyzed together. These results will be presented in the next work. Thank you very much for your suggestions.

Reviewer 3 Report

This paper is on the verge of being rejected, the missing of the most leading groups today in MQL and cryogenics speaks volume!!. Pereira works about Cryogenics, MQL, oil compositions, etc…4-6 works…missed. J Cleaner production was not checked. On the other hand, Angulo work in IJAMT also missed….20 works missed?

  • Anyway, the idea and results are quite interesting that is because Major could be a better consideration, with hard work the paper can be seriously improved.
  • On the other hand…you use weird units and names…if you are using Cutting speed Vc…use the real symbols…x1,x2…how ugly it is.
  • Conclusions: write them as bullets, one per each claim with respect the state of the art.
  • ANOVA was well defined.
  • Al2O3/MoS2 hybrid nanofluid: define how harmful it could be.
  • Figure 1 is interesting, similar to many missed authors…again Pereira….Lacalle…etc

Paper must be absolutely improved, please do it because some results are interesting. see also Drilling of CFRP-Ti6Al4V stacks using CO2-cryogenic cooling, Journal of Manufacturing Processes 64, 58-66

Author Response

RESPONSE TO THE REVIEWER 3

We are very grateful for the reviews provided by the editors and each of the external reviewers of this manuscript. Please see below, our detailed response to comments.

  1. This paper is on the verge of being rejected, the missing of the most leading groups today in MQL and cryogenics speaks volume!!. Pereira works about Cryogenics, MQL, oil compositions, etc…4-6 works…missed. J Cleaner production was not checked. On the other hand, Angulo work in IJAMT also missed….20 works missed?

Anyway, the idea and results are quite interesting that is because Major could be a better consideration, with hard work the paper can be seriously improved.

Answer:

Thank you very much for your valuable comments. This is our shortcoming in the overviewing process. Thanks again to the reviewer for your suggestions on valuable studies. These works have been referenced and cited in detail in red color in the revised manuscript in Refs. 13-18.

  1. On the other hand…you use weird units and names…if you are using Cutting speed Vc…use the real symbols…x1,x2…how ugly it is.

Answer:

The variables x1, x2, and x3 were changed to the real names of nanoparticle concentration, cutting speed, and feed rate in the revised manuscript with red color. Thank you very much

  1. Conclusions: write them as bullets, one per each claim with respect the state of the art.

Answer:

Thank you very much for your comment. The conclusion was rewritten with addition of main contributions by following the reviewer’s comments. The change is put in red color.

  1. ANOVA was well defined.

Answer:

Thank you very much for your comment

  1. Al2O3/MoS2 hybrid nanofluid: define how harmful it could be.

Answer:

Thank you very much for your valuable comment. We also encountered with this problem when doing the experiments. Nanoparticles suspended in cutting fluids in oil mist form will be threats for human health, so the exhaust fans and ventilation system mist are suggested to use to remove the oil mist. The discussion about this problem was added in the revised manuscript in Sec. 3. Discussion with red color.

In our workshop, we already had the exhaust fans and ventilation system to remove the oil mist in order to avoid this problem.

  1. Figure 1 is interesting, similar to many missed authors…again Pereira….Lacalle…etc

Answer:

Thank you very much for your comment

  1. Paper must be absolutely improved, please do it because some results are interesting. see also Drilling of CFRP-Ti6Al4V stacks using CO2-cryogenic cooling, Journal of Manufacturing Processes 64, 58-66

Answer:

Thank you very much. The work has been referenced and cited in detail in red color in the revised manuscript in Ref.18. The results are very interesting, and help us to improve the paper quality

Round 2

Reviewer 3 Report

Good work